# ADDG: An Adaptive Domain Generalization Framework for Prostate Cross-Plane MRI Segmentation

## ABSTRACT

Multi-planar and multi-slice magnetic resonance imaging (MRI) can provide more comprehensive 3D structural information for disease diagnosis. However, compared to multi-source MRI, multi-planar MRI uses almost the same scanning parameters but scans different internal structures. This atypical domain difference may lead to poor performance of traditional domain generalization methods in handling multi-planar MRI, especially when MRI from different planes also comes from different sources. In this paper, we introduce ADDG, an Adaptive Domain Generalization Framework tailored for accurate cross-plane MRI segmentation. ADDG significantly mitigates the impact of information loss caused by slice spacing by incorporating 3D shape constraints of the segmentation target, and better clarifies the feature differences between different planes of data through adaptive data partitioning strategy. Specifically, we propose a mesh deformation-based organ segmentation network to simultaneously delineate the 2D boundary and 3D mask of the prostate, as well as to guide more accurate mesh deformation. We also develop an organ-specific mesh template and employ Loop subdivision for unpooling new vertices to a triangular mesh to guide the mesh deformation task, resulting in smoother organ shapes. Furthermore, we design a flexible meta-learning paradigm that adaptively partitions data domains based on invariant learning, which can learn domain invariant features from multi-source training sets to further enhance the generalization ability of the model. Experimental results show that our approach outperforms several medical image segmentation, single-planar-based 3D shape reconstruction, and domain generalization methods.

## CCS CONCEPTS

• **Computing methodologies → Image segmentation**.

## KEYWORDS

Image Segmentation, Multi-Planar MRI, Domain Generalization, Meta-Learning, Mesh Deformation

## 1 INTRODUCTION

In recent years, magnetic resonance imaging (MRI) has played an increasingly important role in medical diagnosis. However, single-planar MRI is typically used in examinations, and the presence of slice spacing can result in significant information loss, as shown in Figure 1 (a). This can easily lead to inaccurate perception of organ and lesion shapes, as well as occurrences such as missed detection of small lesions [44]. Correspondingly, multi-planar MRI scans the human body through three mutually perpendicular directions, e.g., the axial, coronal, and sagittal planes, as shown in Figure 1 (e). Using multi-planar MRI can significantly reduce information loss caused by slice spacing and effectively improve diagnostic accuracy[38]. But, it requires the radiologists to correlate images from multiple directions, which demands higher expertise and experience, as

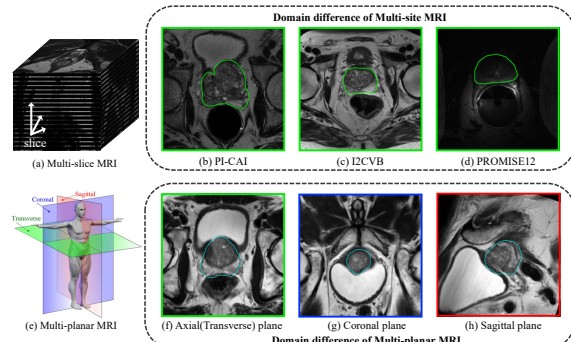

**Figure 1: (a) spatial arrangement of multi-slice MRI; (b-d) MRI of a similar location from the different data sources; (e) MRI's multi-plane scanning mode; (f-h) coronal, sagittal, and axial (transverse) plane MRI. In comparison, multi-source MRI has a similar internal structure but different visual features, while multi-planar MRI is the opposite.**

well as more time. This implies that reviewing multi-planar MRI is more prone to intra-observer and inter-observer errors[46]. In the context of a severe shortage of experienced radiologists and increasing hospital workload, the use of multi-planar MRI seems like a double-edged sword for disease diagnosis.

Fortunately, deep learning has been increasingly applied in intelligent assisted diagnosis. For example, automatic segmentation of organs and lesions from a large number of MRI data can significantly alleviate the workload of radiologists. However, the training of deep learning models often requires a large amount of data with high-quality annotations[1]. Due to restrictions such as medical ethics, medical imaging datasets are typically small in sample size and even incomplete. Most of them also lack or have insufficient (high-quality) annotations. Taking the prostate organ segmentation task as an example, there have been several benchmark datasets that can be used for model training, such as PROMISE12 [25] and Decathlon [37]. But they both have some limitations in terms of multi-plane aspects, i.e., either they contain data and annotations from single-planar MRI or although they include data from multi-planar MR images, annotations are provided only for single-planar MRI. Meanwhile, due to annotating 3D medical image data is time-consuming, collecting a set of annotated multi-planar MRI data is usually impractical. This makes it difficult for researchers to train high-performance models to segment multi-planar MRI.

To achieve accurate organ segmentation on multi-planar MRI without having full-planar annotations, a potentially viable approach is domain generalization[12]. This involves training robust models with annotated plane data and enabling precise organ segmentation on unannotated planes. However, domain generalization is generally used to handle multi-source MRI data, addressing domain shift issues caused by differences in data distribution between

different sources. Those differences are typically manifest in contrast, the field of view (FOV), and other aspects, stemming from scanning parameters, physician preferences, and variations in patient conditions (lesion textures are often more complex), as shown in Figure 1 (b-d). It is worth noting that, for slices at similar locations from different data sources, their internal (organ) structures exhibit higher similarity. Different from multi-source MRI, multi-planar MRI uses almost the same scanning parameters but scans different internal structures, as shown in Figure 1 (f-h). The atypical domain differences between multi-planar MRI almost exclude features like image contrast that traditional deep learning models usually extract. This makes it challenging for domain generalization models to capture sufficient domain differences and correlations, thus failing to meet the requirements of organ segmentation tasks on multi-planar MRI data. To validate the above point, we conducted several experiments using the PROSTATEX-Seg-HiRes dataset [8, 30, 36]. We separately train U-Net, MLDG-based U-Net on a single-planar (axial plane) data, and Voxel2Mesh on multi-planar data. Then, we test them on the sagittal plane data. The results (DSC) are 44.63%, 46.38%, and 52.59%, respectively. While the experiments may not be 100% rigorous, we can still identify that domain-generalization segmentation can alleviate the problem of cross-plane data to some extent. However, compared to the 3D shape reconstruction method based on single-planar images, there still exists a significant performance gap. Unfortunately, the segmentation issues for multi-planar MRI have not received widespread attention from researchers yet, and there is currently a lack of in-depth research dedicated to addressing these issues.

Based on the analysis above, we first introduce a novel domain generalization problem for the task of prostate MRI segmentation: given only the images and annotations from one plane, how can we achieve prostate segmentation in the other two planes? To achieve the objectives above, we propose ADDG, an adaptive domain generalization framework for accurate cross-plane MRI segmentation. Unlike any previous prostate segmentation method or domain generalization approach, ADDG is based on the shape reconstruction method for 3D objects from single-planar images, so that it can estimate the 3D shapes of the prostate from single-planar MRI, enabling it to segment the prostate on unseen plane MR images better. Further, we propose a gradient meta-learning training strategy with adaptive data partitioning. Specifically, ADDG consists of a cross-plane segmentation model based on mesh deformation and a meta-learning training strategy with adaptive data partitioning criterion. For better cross-plane segmentation, we propose a mesh deformation based organ Segmentation network that simultaneously learns the spatial relationships between slices of the 3D image and the 2D image details within each slice. Additionally, we also design a task-specific initialization mesh template and introduce a vertex augmentation method based on Loop subdivision to ensure smoother mesh deformation of the prostate. The meta-learning training strategy with adaptive data partitioning is used to capture more domain shift data partitions from training sets composed of multiple sources (institutions), enabling the model to learn more domain-invariant features and further enhance the generalization ability of the cross-plane segmentation model based on mesh deformation.

Our contributions are as follows:

- We identify a domain generalization problem crucial for prostate three-dimensional modeling, yet previously unexplored, regarding cross-plane MRI segmentation. Then we propose a tailored cross-plane segmentation framework for this problem, which utilizes a 3D object reconstruction model from single-planar images as the backbone.
- We devise a novel meta-learning strategy with adaptive data partitioning for model training. This approach can fully exploit the heterogeneity of multi-domain data, and enable more comprehensive learning of domain-invariant features compared to traditional meta-learning methods.
- We further incorporate a mixed encoding scheme of 2D and 3D features, a 2D slice learning branch, a strong prior prostate mesh template, and Loop subdivision method to enhance the model for achieving more precise prostate segmentation across different planes data.

## 2 RELATED WORK

### 2.1 Shape Estimation from Single Plane Image

3D reconstruction of the object from a fixed viewpoint allows the surface/shape of the object to exhibit good continuity and smoothness in any viewpoint. Mesh is a 3D representation commonly used to model the shape of an object in three dimensions, it consists of polygons, which are essentially discrete representations of a continuous surface. The combinatorial nature of polygons makes it possible to take derivatives in the space of possible meshes for any given surface. As a result, mesh processing and optimization techniques have difficulty utilizing the modular gradient descent component of modern optimization frameworks. To circumvent this problem, Deformation-based Mesh Generation (DEMG) methods[19] has attracted more attention. A salient feature of DEMG methods is the need for an initial mesh such as a spherical or elliptical template mesh. Due to the presence of an initial mesh, this type of approach reduces the difficulty of mesh generation to some extent. The neural network only needs to predict the positions of the vertices because the connection relationship between the vertices already exists.

Wang[41] first proposed a deep learning based approach to extract 3D triangular meshes of object from singe RGB image, Wen[42] et al. introduced RGB images with different viewpoints to make the generated 3D shapes more accurate, considering that DEMG methods can only generate meshes with topology similar to the initial mesh. For this reason, Ben[2] introduced a faceted pruning mechanism, which iteratively adjusts the topology of the mesh through faceted pruning operations while maintaining the main attributes of the template, i.e., visually appealing and uniform mesh connections. Admittedly, the faceted pruning mechanism allows the method to be adapted to mesh generation with more complex topologies.

In the context of medical imaging, DEMG methods has been widely noticed and used in the reconstruction of 3D organs in question, taking into account this property of the sphere-like shape of the organs. Wickramasinghe[43] firstly proposed an end-to-end trainable deep learning based architecture that takes an image volume as input and outputs a 3D surface mesh, Kong[17, 18] based on Voxel2Mesh, realized 3D shape modeling of heart from multiplane cine MRI and 3D modeling of cardiac process from movie

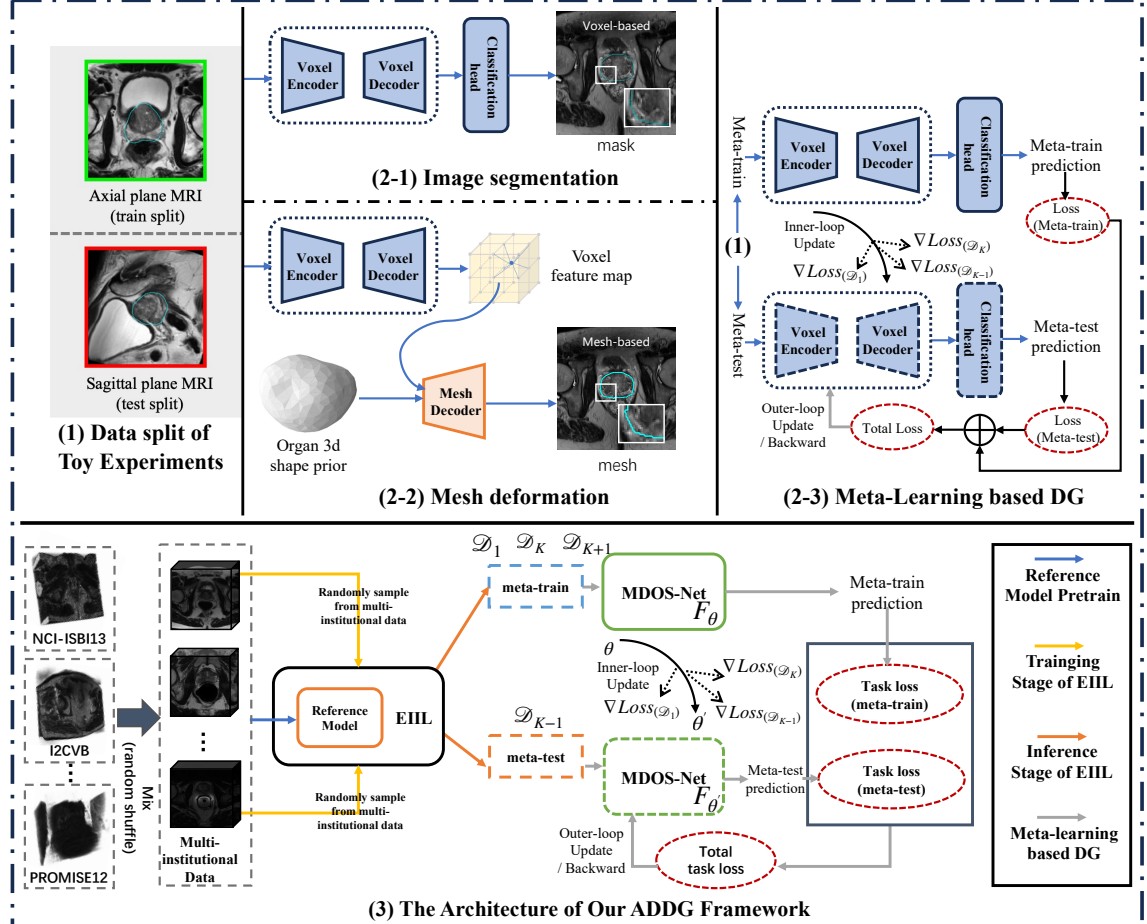

Figure 2: Illustration of mesh deformation method (2-2), meta-learning-based DG (2-3), image segmentation (2-1) methods, and our ADDG framework (3) respectively.

cine MRI, respectively; Bongratz[5] modified Voxel2Mesh, realized the 3D shape modeling of complex brain structures (i.e. cerebral cortex and sulcal gyrus).

## 2.2 Domain Generalization for Medical Image

The goal of domain generalization is to train a model using data from a single or multiple related but distinct domains in such a way that the model can generalize well to any out-of-distribution (OOD) unseen test domains. The significance of domain generalization is particularly high in medical image analysis, data are often non-independently distributed (non-iid), which is also known as data heterogeneity in the medical image analysis. Appearance variability in medical imaging refers to differences and inconsistencies typically manifest during the data acquisition step[31], which is called data heterogeneity. This variability may arise externally from using different modalities, protocols, scanner types, and patient populations across multiple healthcare facilities, while internal variability may also occur within a controlled setting (e.g., same scanner or healthcare facility) due to factors such as hardware aging, software parameter variations, and human error (e.g., human motion).

The domain generalization problem has been extensively studied in the area of medical image analysis, Liu[27] proposed a meta-learning-based domain generalization method for prostate segmentation, which pays attention to the distribution shift problem in cross-site pelvis axial plane MRI. Li[23] propose to learn an invariant feature through variational encoding with linear-dependency regularization term to equip the model with better generalization capability, which is evaluated on skin lesion classification and cord gray matter segmentation via cross-site dermoscopy images and spinal MRI. Liu[26] designed a boundary-oriented episodic learning paradigm to adapt the distribution shift between data from different sources and validated on multi-site retinal fundus images and prostate axial plane MRI. Kang[16] proposed a method with perturbs the training distribution by mixing the styles of training images and a dual-branch invariant content Synergistic learning strategy, which solving the domain generalization problem by controlling the inductive bias and showed the effectiveness on multi-site retinal fundus images and prostate axial plane MRI dataset. Kamraoui[15]proposed a framework for Multiple Sclerosis lesion segmentation from brain fMRI, which is designed for domain generalization problem.

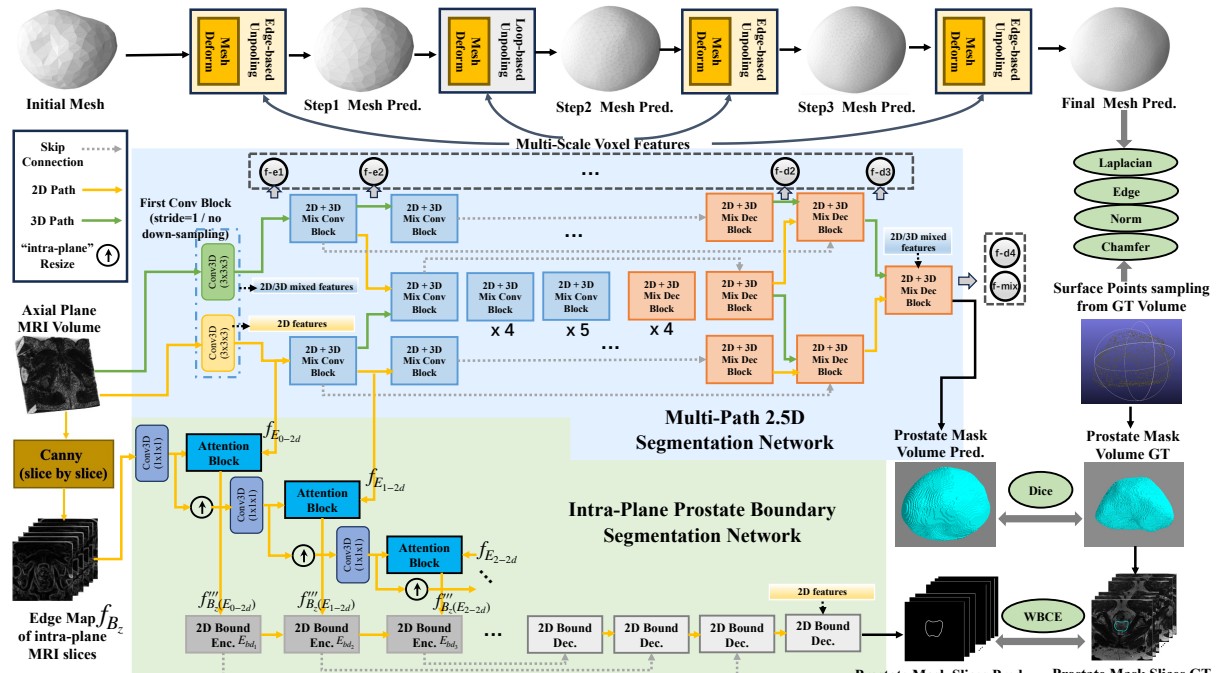

**Figure 3: Architecture of the proposed MDOS-Net.**

To the best of our knowledge, all of domain generalization methods [16, 23, 26, 27] for medical image analysis only focus on the distribution shift problem due to differences in image scans, patient demography, etc. under the same plane data, while neglecting the fact that the data from different planes can also cause the distribution shift problem.

## 3 METHODS

### 3.1 Overview of the ADDG Framework

The proposed ADDG framework aims to segment the prostate gland from unseen plane MRI, as shown in Figure 2. ADDG consists of a **M**esh **D**eformation-based **O**rgan **S**egmentation network (**MDOS-Net**) and a **D**ata-**P**artitioning-**F**ree **M**eta **L**earning (**DPF-ML**) training strategy. Specifically, the MDOS-Net aims to make a better estimation of the prostate shape from single-planar MRI, and the DPF-ML training strategy aims to capture more domain invariant representations by fully utilizing the axial plane MRI from multiple institutions.

### 3.2 Mesh Deformation based Organ Segmentation Network (MDOS-Net)

In order to better utilize 3D contextual information to guide cross-plane segmentation, we first use axial plane MRI to produce a 3D mesh model in the anatomical space coordinate, and then convert the 3D mesh to voxel segmentation mask via the rasterization algorithm. The basic idea is inspired by Voxel2Mesh [43], which has a strong capability to produce reasonable shapes even from single-planar MRI. However, Voxel2Mesh does not take into account the anisotropy presented in 3D MRI, i.e., the slice spacing is usually much larger than the actual distance represented by a single pixel within each slice. Anisotropic MRI could lead to the model

overfitting to inter-slice noise, resulting in a decrease in model performance [10]. Thus, we propose MDOS-Net that can better utilize the rich information in 3D MRI to achieve better cross-plane segmentation. As shown in Figure 3, this backbone network consists of three parts: 1) a Multi-Path 2.5D Segmentation Network to handle data with varying degrees of anisotropy; 2) an Intra-Plane Prostate Boundary Segmentation Network to leverage high-resolution 2D image slices from anisotropic MRI to learn 2D edge information of target organs; and 3) a 3D triangular mesh deformation branch to produce the shape of target organ precisely.

#### 3.2.1 *Multi-Path 2.5D Segmentation Network.* Considering the continuity of organ appearance in 3D MRI, treating a 3D segmentation task as a stack of consecutive 2D tasks may lead the model to overlook contextual information of the segmentation targets, which can easily result in under- or over-segmentation [14]. This issue can be further exacerbated in cross-plane segmentation. One the other hand, most current methods typically do not specifically address the issue of anisotropy in 3D medical images [10, 45], and the inconsistent slice spacing between different scans makes it difficult for anisotropic convolutional kernels to robustly handle unpredictable data.

Our multi-path 2.5D segmentation network is inspired by MNet [10], which simultaneously combines multiple representation processes involving 2D, 3D, and combinations of two features adaptively. In particular, apart from the initial encoder that includes one layer each of 2D and 3D convolutions, the remaining encoder and decoder blocks incorporate skip connections [13], as illustrated in Figure 3 (more details in supplementary materials ). Furthermore, we incorporate channel attention [32] into the decoder, and employ group normalization, which is better suited for small batch sizes (due to limitations in GPU memory capacity, i.e., 24GB RAM per

GPU, we set the batch size to 1). Unlike pre-selected feature maps, we adopt multi-scale feature training strategy[5], which ensures that the network learns rich multi-scale features during the training process. We also simultaneously feed multi-scale voxel and 2D feature maps from both the encoder and decoder into the mesh deformation branch, as shown in Figure 3.

### 3.2.2 Intra-Plane Prostate Boundary Segmentation.

Due to the anisotropy of 3D MRI is reflected in intra-slice with high resolution and inter-slice with low resolution, we introduce a 2D boundary detection as an auxiliary task by designing an intra-plane prostate boundary segmentation network as shown in 3, which aims to enhance the details of intra-plane boundaries in multi-scale feature maps within the backbone network, enriching the feature representative ability.

This branch processes the 3D image in a slice-by-slice manner, utilizing the first derivative operator in both horizontal and vertical directions to extract the low-level boundary information while filtering the trivial boundary irrelevant information in each intra-slice. However, these low-level boundaries often contain excessive noise unrelated to the 2D prostate region. To address this issue, we adopt an attention mechanism [32] to filter out the noise and aggregate image features from both the segmentation backbone and the low-level boundaries. Since the 2D boundary within a slice can be regarded as a 2D elongated tubular structure, we replace the normal 2D convolution with a 2D dynamic snake convolution [35] in the branch.

Specifically, we extract the 2D feature map $f_{E_{i-2d}}$ from encode stage $i$ of the multi-path 2.5D segmentation network, with all potential edge maps $f_B$ extracted slice by slice from image by Canny edge detector [6]. In the step where the Canny edge detector finds the gradient size and direction, we use two $3 \times 3$ parameter fixed convolutions (with stride 1). Then we fuse $f_B$ and $f_{E_{i-2d}}$ through the following manner: the edge map $f_{B_z}$ of each slice indexed by the z-axis, we resize $f_{B_z}$ by bilinear interpolation for each slice $j$ to get $f'_{B_j(E_{i-2d})}$, so that it matches the dimension of (W, H) of $f_{E_{i-2d}}$; then the channel dimension of $f'_{B_j(E_{i-2d})}$ is matched to $f_{E_{i-2d}}$ at the current scale by $1 \times 1 \times 1$ convolution, to obtain $f''_{B_j(E_{i-2d})}$. Then $f_{E_{i-2d}}$ is subjected to aggregate by attention with $f''_{B_j(E_{i-2d})}$, resulting potential contour detail feature map, $f'''_{B_j(E_{i-2d})}$, is used as the input to the 2D boundary encoder block $E_{bd_k}$ based on the 2D dynamic snake convolution:

$$f'''_{B_j(E_{i-2d})} = \sigma_2 \left( W_\varphi \left( \sigma_1 \left( W_{f''_{B_j}} \cdot f''_{B_j} + W_{f_{E_{i-2d}}} \cdot f_{E_{i-2d}} \right) \right) \right) \cdot f_{E_{i-2d}},$$
(1)

where $W_{f''_{B_j}}$, $W_{f_{E_{i-2d}}}$, and $W_\varphi$ denote the linear transform using $1 \times 1 \times 1$ convolutions, $\sigma_1$ and $\sigma_2$ represent ReLU and Sigmoid activation functions, respectively.

### 3.2.3 Organ-Specific Mesh Template.

To further improve the accuracy of mesh deformation, we create an organ-specific mesh template for the deformation stage. Specifically, we utilize 3D segmentation mask from PROSTATEx-Seg-HiRes [30, 36], which annotates the prostate for all three planars. Considering the generalizable shape of the prostate would bring the appropriate mesh initialization, we refer to the clinical TNM staging provided by the

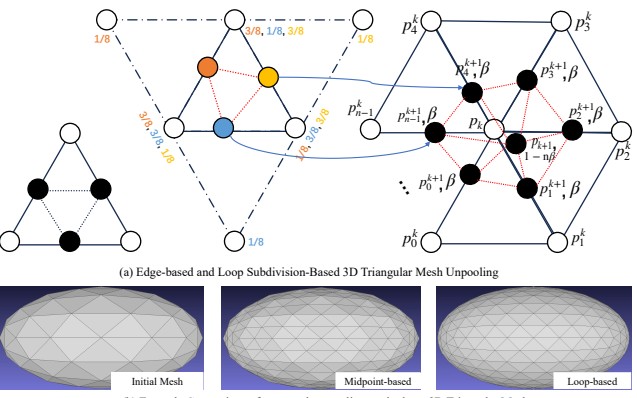

(a) Edge-based and Loop Subdivision-Based 3D Triangular Mesh Unpooling

(b) Example Comparison of two graph unpooling methods on 3D Triangular Mesh

**Figure 4: (a) Vertices in orange, yellow, blue, black and red dashed edges are added in the unpooling layer. The blue curve represents the new vertex of a face. (b) The midpoint based unpooling leads to unsmooth surface topology, while the loop-based unpooling remains smoother.**

PROSTATEx dataset [8, 11, 24] and select a relatively healthy case with the lowest degree of staging as our template data. Following Lewiner's marching cubes algorithm [21], we first extract an ico sphere and then apply Laplacian smoothing [39] with 50 iterations to obtain a prostate mesh consisting of 51,268 vertices and 102,532 faces. Due to limitations in GPU memory capacity (24GB per card), we adopt triangle mesh simplification while maximizing the retention of surface topography. As a result, we obtain a prostate mesh with 388 vertices and 762 faces.

### 3.2.4 Loop Subdivision Based Vertex Unpooling.

The current methods for reconstructing 3D organs [5, 17, 18, 43] usually use midpoint-based unpooling from Pixel2Mesh [41] to add the new vertices for mesh. While the midpoint-based unpooling does not increase the vertex degree, it adds a certain level of angularity to the mesh shape. To achieve a smoother mesh, we adopt Loop subdivision [29] for vertices unpooling. Unlike midpoint subdivision, which only adds new vertices along each edge and its two existing vertices, Loop subdivision not only add new vertices by more neighbor vertices, but also updates the position of the old vertices by merging its position with those of its neighboring vertices. As shown in Figure 5 the subdivided mesh by Loop subdivision appears smoother compared to the subdivided mesh by midpoint.

As illustrated in Figure 5, $p_k$ is defined as an origin vertex, where $k$ is the subdivision steps. $p_0$ is defined as the control vertex of the mesh. Then we can obtain $p_\infty$ from $p_0$ after $\infty$ steps as $p^0 \rightarrow p^1 \rightarrow \cdots \rightarrow p^\infty$. Suppose $p_k$ has $n$ neighbor vertices and $p_k^i$, $i \in \{0, 1, \ldots, n-1\}$, then the valence of $p_k$ is $n$. For each edge between $p_k$ and $p_k^i$, a new vertex $p_{k+1}$ was created.

$$f\left(p_i^{k+1}\right) = \frac{3f\left(p^k\right) + 3f\left(p_i^k\right) + f\left(p_{i-1}^k\right) + f\left(p_{i+1}^k\right)}{8}.$$
(2)

$$f\left(p^{k+1}\right) = (1 - n\beta)f\left(p^k\right) + \beta\left(f\left(p_0^k + \cdots + f\left(p_{n-1}^k\right)\right)\right),$$
(3)

where Eq. (2) updates the position of old vertex $p_k$ to $p_{k+1}$, Eq. (3) creates the new vertex $p_{k+1}$ between $p_k$ and $p_k^i$, $\beta$ is a constant:

$$\beta = \frac{1}{n}\left(\frac{5}{8} - \frac{3 + 2(\cos 2\pi/n)^2}{64}\right). \quad (4)$$

*3.2.5* ***Loss Function****.* Since ADDG is a multi-task framework (i.e. including 3D voxel segmentation, 2D intra-plane boundary segmentation and 3D triangular mesh prediction), we define a comprehensive loss function to optimize these tasks jointly. The total loss function consists of $\mathcal{L}_{voxel}$ for the Voxel Segmentation part and $\mathcal{L}_{mesh}$ for the mesh deformation part:

$$\mathcal{L}_{total} = \mathcal{L}_{voxel} + \mathcal{L}_{mesh}. \quad (5)$$

**Voxel Loss** We use the Sørensen–Dice Loss $\mathcal{L}_{Dice}$ for the multi-path 2.5D segmentation network:

$$\mathcal{L}_{\text{Dice}} = 1 - \frac{2\sum_{i=1}^{N}(y_i * \hat{y}_i)}{\sum_{i=1}^{N}(y_i + \hat{y}_i)}, \quad (6)$$

where $N$ denotes the total number of voxels, $y_i$ and $\hat{y}_i$ denote the ground truth and the prediction for the voxel $i$, respectively. For intra-plane prostate boundary segmentation network, considering the class imbalance problem between the foreground and background pixels in the boundary segmentation task, we use the weighted binary cross-entropy loss $\mathcal{L}_{\text{WBCE}}$:

$$\mathcal{L}_{\text{WBCE}} = -\frac{1}{N}\sum_{i=1}^{N}\left(w_f y_i \log(\hat{y}_i) + w_b(1 - y_i)\log(1 - \hat{y}_i)\right), \quad (7)$$

where $w_f$ and $w_b$ denote the hyper-parameters of weighting factor for foreground voxels and background voxels, respectively. The overall voxel loss function $\mathcal{L}_{voxel}$,

$$\mathcal{L}_{\text{voxel}} = \lambda_1 \mathcal{L}_{Dice} + \lambda_2 \mathcal{L}_{\text{WBCE}}, \quad (8)$$

where $\lambda_i$ denotes hyper-parameters that determine the relative significance of each loss term.

**Mesh Loss** In the mesh deformation step, we use the loss term from [43]: the chamfer distance loss $\mathcal{L}_{cd}$ to measure geometry consistency; three geometry regularization loss, including Laplacian smoothing of the displacement fields loss $\mathcal{L}_{lap}$ to prevent self-intersection of faces, the intra-mesh normal consistency loss $\mathcal{L}_n$ which measures the consistency of normals within the mesh, and the edge length loss $\mathcal{L}_{el}$ which penalizes vertices that are too far apart and edges that are too long. The overall mesh loss $\mathcal{L}_{mesh}$ is:

$$\mathcal{L}_{\text{mesh}} = \lambda_3 \mathcal{L}_{cd} + \lambda_4 \mathcal{L}_n + \lambda_5 \mathcal{L}_{el} + \lambda_6 \mathcal{L}_{lap}, \quad (9)$$

where $\lambda_i$ denotes hyper-parameters that determine the relative significance of each loss term.

## 3.3 Data-Partitioning-Free Meta-Learning

Due to the persistent issue of small sample sizes in public datasets for prostate MRI segmentation, we aggregate multiple datasets into a larger training set to improve the generalization performance of the proposed model. While the datasets were collected from different institutions, simply training with aggregated data will make the model performance degradation [28]. Moreover, data heterogeneity not only occurs within training sets containing multi-site data, but also exists between the training and the test set, which makes the trained model perform poorly on the test set. To this end, based on MLDG [22], which has been widely used in domain generalization of medical image segmentation tasks due to its plug-and-play

---

**Algorithm 1:** Pseudocode of DPF-ML

**Input:** Dataset $\mathcal{S} = \{x_i, y_i\}$, Loss $\ell = \mathcal{L}_{Dice}$, Loss $\mathcal{F} = \mathcal{G} = \mathcal{L}_{total}$,
    Training steps $N_{steps}$ for meta-train/meta-test split inference
**Init:** reference model $\Phi$ for EI objective, MDOS-Net with parameters $\Theta$
**Objective loss function of Invariant Risk Minimization (IRM):**
**def** $\tilde{R}^e(\Phi, \mathbf{q})(x)$:
    **return** $\frac{1}{\sum_{i'}\mathbf{q}_{i'}(e)}\sum_i \mathbf{q}_i(e)\ell(\Phi(x_i), y_i)$

Randomly init. $\mathbf{q} \in [0,1]^N$ data split posterior ($\mathbf{q}_i(e) := q(e \mid x_i, y_i)$)
**for** $n \in 1 \ldots N_{steps}$ **do**
    **Aggregate variances of $\Phi$ across soft domains:**
        SoftVari $= \sum_{e \in \{1,2\}}\|\nabla_{\tilde{w}}\tilde{R}^e(\bar{w} \circ \Phi, \mathbf{q})\|$
    **Maximize the EI objective by minimizing the Loss:**
        Loss $= -1 \cdot$ SoftVari
    **Update domain posterior q:** $\mathbf{q} \leftarrow$ OptimUpdate $(\mathbf{q}, \nabla_{\mathbf{q}}\text{Loss})$
**end for**
$\hat{\mathbf{q}} \sim$ Bernoulli$(\mathbf{q})$
$\overline{\mathcal{S}} \leftarrow \{x_i, y_i \mid \hat{\mathbf{q}}_i = 1\}$, $\check{\mathcal{S}} \leftarrow \{x_i, y_i \mid \hat{\mathbf{q}}_i = 0\}$
**return** *meta-train split* $\overline{\mathcal{S}}$, *meta-test split* $\check{\mathcal{S}}$
**for** *epoch* in *Epochs* **do**
    **for** *iters* in *maxlen*$(\overline{\mathcal{S}}, \check{\mathcal{S}})$ **do**
        **Sample:** $\bar{s} \in \overline{\mathcal{S}}$, $\check{s} \in \check{\mathcal{S}}$
        **Meta-Train Step:** Compute gradients $\nabla_\Theta = \mathcal{F}'_\Theta(\overline{\mathcal{S}}; \Theta)$
        Update parameters $\Theta' = \Theta - \alpha\nabla_\Theta$
        **Meta-Test Step:** Compute loss $\mathcal{G}(\check{\mathcal{S}}; \Theta')$
        **Meta-Optimization Step:** Update $\Theta$
        $\Theta = \Theta - \gamma\frac{\partial\left(\mathcal{F}(\overline{\mathcal{S}};\Theta) + \beta\mathcal{G}(\check{\mathcal{S}};\Theta - \alpha\nabla_\Theta)\right)}{\partial\Theta}$
    **end for**
**end for**
**end procedure**

---

convenience, and attempts to solve the issue of model performance degradation when training and test data were acquired from different sources, by utilizing the training data which aggregate from different institutions.

The core idea of MLDG-based methods is to learn the domain invariant representation by simulating the distribution shift from the multi-source data. Although powerful, it requires a discrete set of domain labels for partitioning the meta-train and meta-test set, each corresponding to specific data distribution, during the training stage. As a result, it seems not appropriate to directly apply this method to the domain generalization problem where pre-defined domain labels are unavailable, especially for medical images. Most domain generalization methods try to allocate the domain labels by simply treating a dataset from one source as a domain. This partitioning criterion overlooks the diversity that exists within the dataset. Taking the prostate MRI dataset for segmentation as an example, these data may come from various medical institutions, with different field strengths (1.5T, 3T), scanning equipment (Siemens, GE, Phillips), scanning parameters (e.g., whether coils are placed during scanning, and where they are placed), and variations in the patient's conditions. This makes it challenging to get an appropriate data partition for meta-train and meta-test based on this complicated information. Additionally, the atypical domain differences in multi-plane MRI further exacerbate this issue.

Inspired by invariant learning method EIIL[9], we propose the **Data-Patitioning-Free Meta-Learning (DPF-ML)**, to break the neglect of the diverse potential data distributions contained in multi-institutional medical images due to the fixed data partitioning criterion (neither predefines different domains manually nor

Table 1: Comparison of ADDG with related works

| Method | Coronal Plane MRI | | | | Sagittal Plane MRI | | | |
|---|---|---|---|---|---|---|---|---|
| | Dice (%, ↑) | HD95 (mm, ↓) | aRVD (%, ↓) | ASSD (mm, ↓) | Dice (%, ↑) | HD95 (mm, ↓) | aRVD (%, ↓) | ASSD (mm, ↓) |
| 3D U-Net 2016 [7] | 59.73 ± 4.25 | 21.58 ± 6.72 | 23.45 ± 7.70 | 7.91 ± 2.56 | 48.51 ± 6.02 | 28.01 ± 6.53 | 29.71 ± 7.32 | 12.09 ± 2.49 |
| SAM-Med3D 2023 [40] | 60.61 ± 3.71 | 21.14 ± 6.62 | 21.72 ± 7.24 | 7.76 ± 2.81 | 49.92 ± 6.75 | 27.73 ± 7.07 | 29.53 ± 6.56 | 12.35 ± 2.88 |
| SAML 2020 [27] | 61.88 ± 3.95 | 19.60 ± 5.81 | 20.11 ± 6.86 | 6.92 ± 2.55 | 51.05 ± 6.23 | 26.81 ± 6.61 | 27.95 ± 6.75 | 12.11 ± 2.54 |
| IB U-Nets 2022 [3] | 62.35 ± 3.20 | 17.55 ± 5.75 | 18.12 ± 5.01 | 6.70 ± 2.19 | 51.42 ± 6.81 | 26.09 ± 6.95 | 26.82 ± 6.99 | 11.26 ± 2.75 |
| nnUNet 2021 [14] | 63.22 ± 3.03 | 16.97 ± 5.97 | 17.94 ± 6.05 | 6.48 ± 2.28 | 52.15 ± 6.60 | 25.96 ± 7.27 | 26.64 ± 6.81 | 10.78 ± 2.49 |
| MNet 2022 [10] | 63.95 ± 3.92 | 17.15 ± 6.01 | 17.79 ± 6.23 | 6.54 ± 2.35 | 51.93 ± 7.11 | 26.60 ± 6.92 | 27.05 ± 7.53 | 11.05 ± 2.64 |
| Voxel2Mesh 2020 [43] | 65.10 ± 2.75 | 15.71 ± 4.92 | 16.05 ± 4.63 | 5.57 ± 1.91 | 55.52 ± 4.12 | 24.72 ± 3.75 | 25.15 ± 4.10 | 9.88 ± 1.43 |
| Vox2Cortex 2022 [5] | 66.95 ± 3.67 | 13.35 ± 5.23 | 13.07 ± 5.45 | 5.25 ± 1.99 | 56.15 ± 4.28 | 24.41 ± 4.63 | 25.19 ± 4.82 | 9.76 ± 1.79 |
| **ADDG (Ours)** | **70.03 ± 4.91** | **12.12 ± 4.75** | **12.58 ± 5.22** | **4.63 ± 1.85** | **57.91 ± 5.72** | **23.50 ± 4.38** | **24.07 ± 5.09** | **8.97 ± 1.67** |

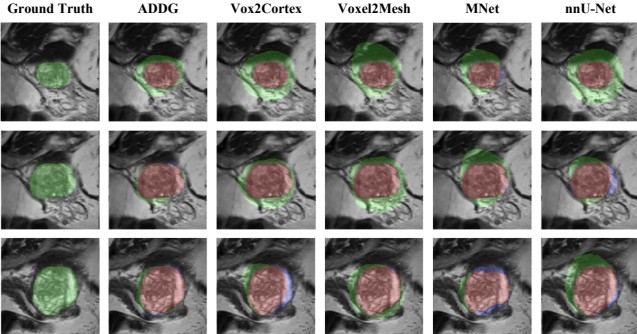

Figure 5: Visual test on the sagittal plane, where red mask denotes true positive, green mask in column 1-5 denotes false positive, and purple mask denotes false negative.

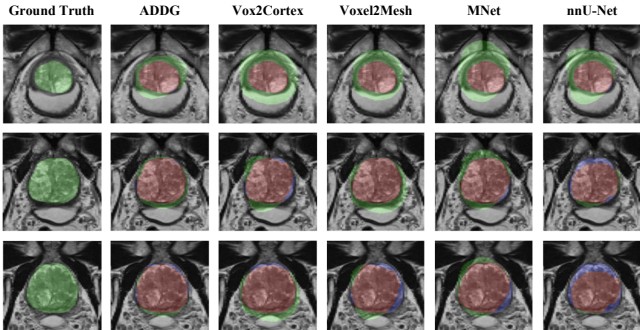

Figure 6: Visual test on coronal plane, where red mask denotes true positive, green mask in column 1-5 denotes false positive and purple mask denotes false negative.

randomly samples data from a multi-site training set to obtain data partition for meta-train and meta-test split) in MLDG, then the data partitioning process of the MLDG is turned into a learnable process by modeling more potential data distribution drifts based on invariant learning theory, which allows MLDG to learn a wider range of domain-agnostic features.

As illustrated in Algorithm 1, the proposed DPF-ML includes three stages: 1) reference model obtain; 2) domain inference via invariant learning and 3) meta-learning. In step 1, we first train a model $\Theta$ using Empirical Risk Minimization (ERM), which has the same structure as the "Multi-Path 2.5D Segmentation Network"; In step 2, our goal is to find partitions (data split) from the multi-site training data that maximally violate the invariance principle. Note that maximizing the invariance principle violation would be equivalent to the worst-case scenario on which we would like to train our model, which corresponds to maximizing the regularization term $\frac{1}{\sum_{i'} \mathbf{q}_{i'}(e)} \sum_i \mathbf{q}_i(e)\ell\left(\Phi\left(x_i\right), y_i\right)$ in Algorithm 1. In each iteration of step 2, we update the posterior probability $\mathbf{q}$ of the meta-train and meta-test split, and after $N_{steps}$ iterations, we obtain the meta-train and meta-test sets. Finally, in step 3, we could use the data partition obtained from the previous step.

## 4 EXPERIMENTS

### 4.1 Data and Experiment Settings

*4.1.1 Dataset and Pre-processing.* We utilize axial plane T2w MR images with corresponding segmentation masks (181 patients in

total) from PROMISE12 [25], NCI-ISBI13 [4, 8], I2CVB [20] and De-cathlon [37] for training and validation, and the coronal and sagittal plane MR images (66 patients for each plane) from PROSTATEx-Seg-HiRes [8, 30, 36] for test.

We first conduct the bias field correction and anisotropic dif-fusion noise filter from [33]. For the training and validation data, we first crop or pad them to the size of $176 \times 176 \times 64$ and then resize them to $128 \times 128 \times 128$. For the test data, following [30], we first resample the axial plane MRI (only for orthogonal plane align-ment references), coronal plane MRI, and sagittal plane MRI into a common coordinate system. Next, we crop the images to create a volume that encompasses all three plane MR images. Then, we further crop or resize the image volume in the size of $128\times128\times128$. We also crop the gray values to the 1st and 99th percentiles, then normalize them to [0,1] using Z-score normalization.

*4.1.2 Experiment Settings.* To better evaluate the generalization performance of our proposed model, we utilize *Training-domain validation set* strategy [12], which has been widely used in domain generalization problem. Specifically, the images from each dataset are split into two parts for training and validation, respectively. All training parts are combined for training while all validation parts are combined for selecting the best model. All experiments use five-fold cross-validation.

*4.1.3 Evaluation Metrics.* We present quantitative measures in terms of the Dice Similarity (Dice), the 95-percentile Hausdorff distance (95HD), the absolute relative volume difference (aRVD), and the average symmetric surface distance (ASSD). Meanwhile, we

**Table 2: Ablation study results**

| Method | | | | | | | | Coronal Plane MRI | | | Sagittal Plane MRI | | |
|---|---|---|---|---|---|---|---|---|---|---|---|---|---|
| V2M | MLDG | DPF-ML | AnisoV1 | AnisoV2 | Ellip | PC | Loop | Dice (%, ↑) | HD95 (mm, ↓) | aRVD (%, ↓) | Dice (%, ↑) | HD95 (mm, ↓) | aRVD (%, ↓) |
| ✓ | | | | | | | | 65.10 ± 2.75 | 15.71 ± 4.92 | 16.05 ± 4.63 | 55.52 ± 4.12 | 24.72 ± 3.75 | 25.15 ± 4.10 |
| ✓ | ✓ | | | | | | | 65.48 ± 4.11 | 16.15 ± 5.61 | 16.88 ± 5.50 | 55.76 ± 4.95 | 25.03 ± 4.52 | 25.56 ± 4.75 |
| ✓ | | ✓ | | | | | | 66.05 ± 3.85 | 15.72 ± 5.43 | 16.64 ± 5.65 | 56.15 ± 4.63 | 24.85 ± 4.49 | 25.37 ± 4.40 |
| ✓ | | | ✓ | | | | | 66.86 ± 4.23 | 15.50 ± 5.10 | 15.79 ± 4.91 | 56.91 ± 5.16 | 24.41 ± 4.23 | 24.97 ± 4.31 |
| ✓ | | | | ✓ | | | | 68.02 ± 4.61 | 15.03 ± 5.65 | 15.45 ± 5.11 | 57.42 ± 5.40 | 24.08 ± 4.59 | 24.54 ± 4.35 |
| ✓ | | | | | ✓ | | | 65.23 ± 2.78 | 15.64 ± 5.01 | 16.10 ± 4.48 | 55.60 ± 4.08 | 24.58 ± 3.81 | 25.13 ± 4.12 |
| ✓ | | | | | | ✓ | | 65.90 ± 2.91 | 15.21 ± 5.09 | 15.76 ± 4.92 | 55.96 ± 4.27 | 24.33 ± 3.90 | 24.94 ± 4.22 |
| ✓ | | | | | | | ✓ | 65.51 ± 3.15 | 15.09 ± 4.85 | 15.35 ± 5.01 | 56.04 ± 4.34 | 23.95 ± 3.65 | 24.69 ± 4.25 |
| ✓ | | ✓ | | ✓ | | | | 68.97 ± 4.79 | 14.11 ± 5.35 | 14.72 ± 5.29 | 56.98 ± 5.25 | 24.15 ± 4.72 | 24.50 ± 4.64 |
| ✓ | | ✓ | | ✓ | | ✓ | | 69.55 ± 4.83 | 13.33 ± 5.06 | 13.50 ± 5.41 | 57.53 ± 5.10 | 23.92 ± 4.66 | 24.31 ± 5.05 |
| ✓ | | ✓ | | ✓ | | ✓ | ✓ | **70.03 ± 4.91** | **12.12 ± 4.75** | **12.58 ± 5.22** | **57.91 ± 5.72** | **23.50 ± 4.38** | **24.07 ± 5.09** |

use the rasterization algorithm [34] to convert mesh to segmentation masks, then compute the metrics above with the ground-truth volume masks to evaluate the segmentation performance.

## 4.2 Comparison with Related Works

We compare our approach with several related works for medical image segmentation, domain generalization and organ reconstruction tasks, including 3D U-Net [7], IB U-Nets [3], nnUNet [14], SAM-Med3D [40], SAML [27], Vox2Cortex [5], Voxel2Mesh [43] and MNet [10]. For the best understanding of vision, we overlay the rasterization results of mesh predictions on corresponding coronal and sagittal plane T2w MRI in order to compare the surface reconstruction methods with some other voxel semantic segmentation methods and ground truth segmentation masks.

As presented in Table 1, compared to other methods, our approach achieves the best performance in all evaluation metrics, with improvements of 3.08%-10.3% (dice, coronal plane) and 1.76%-9.4% (dice, sagittal plane). It is worth noting that our approach achieves a generalized segmentation accuracy of 70% (dice) for the first time on the coronal plane. We visualize the segmentation results of the best five methods on the coronal plane MRI and the sagittal plane MRI, as shown in Figure 5 and Figure 6. We can observe that our approach has smoother and more accurate segmentation results compared to other methods. In addition, we found that the sagittal plane is less effective than the coronal plane. Combined with previous visualizations of multi-planar MRI scan data in Figure 1, we argue that there is a more obvious difference between sagittal plane MRI and the other two plane MR images.

## 4.3 Ablation Study

We analyze the contribution and effectiveness of individual design choices in ADDG. We replace our key blocks and learning strategy with the Voxel2Mesh (V2M) method as follows: **MLDG**: we incorporate the gradient-based meta-learning strategy to Voxel2Mesh, which is expected to make a better adaptation to the domain generalization problem. **DPF-ML**: an invariant learning-based approach for data partitioning during model training. **AnisoV1**: in response to the inherent anisotropy of prostate MRI, we have introduced a multi-path 2.5D segmentation network based on MNet. Furthermore, inspired by Vox2Cortex, we intricately concatenated voxel

feature maps of varying scales stemming from both the encoder and decoder and subsequently fed them into diverse stages of mesh deformation, respectively. **AnisoV2**: based on "AnisoV1", we design an intra-plane prostate boundary segmentation network that enhances the feature learning of the segmentation model by learning the 2D prostate boundaries of the intra-plane. **Ellip**: start the mesh deformation from an ellipsoidal template instead of the icosahedron sphere template. We use an ellipsoidal template with the same radius ratios of the x,y, and z axes as the voxel ratios of the prostate. **PC**: start the mesh deformation from our organ-specific mesh template instead of the icosahedron sphere template. **Loop**: in the mesh deformation step, we replace edge-based unpooling [41] in the intermediate step with Loop Subdivision [29] to add new vertices to the triangular mesh before each deformation step. As presented in Table 2, we can observe that each module has made a contribution to the model, and the ADDG (V2M + DPF-ML + AnisoV2 + PC + Loop) method achieves the best performance.

## CONCLUSION

In this paper, we identify a domain generalization problem different from that caused by previous MRI data heterogeneity and define it as the generalization problem of the cross-plane MRI segmentation task. To this end, unlike previous voxel segmentation methods, we introduce a 3D object surface reconstruction method based on single-planar images, which provides a better estimation of the 3D organ shape and breaks the discontinuity of the segmentation results due to the lower inter-slice resolution of the single-planar MRI. Based on this, we design a multi-path 2.5D segmentation branch to handle data with varying degrees of anisotropy, and an intra-plane prostate boundary segmentation branch to leverage high-resolution 2D image slices from anisotropic MRI to learn 2D edge information of target organs. We also propose an adaptive meta-learning strategy that makes the data partitioning process of domain generalization learnable, enabling the model to learn more domain-invariant features. We evaluate our proposed approach using several benchmark datasets of multi-planar prostate MRI. The results indicate that our approach achieves smoother and more accurate prostate segmentation, outperforming other comparative methods, including several image segmentation, domain generalization, and organ reconstruction techniques.

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
