# OpenReview forum: "ADDG: An Adaptive Domain Generalization Framework for Cross-Plane MRI Segmentation"
_acmmm.org/ACMMM/2024/Conference — MM2024 Poster_

### Official Review · Reviewer_yGRa · 2024-04-28

**Rating:** 3
**Confidence:** 3

**Summary:**

This paper is about cross-plane segmentation of prostate in MRI by presenting a method that one only needs annotation on one plane and then the method can segment the prostate on the other two perpendicular planes. In doing this, the paper proposes a series of operations, including mesh representation of a prostate, smoothing of the mesh, and cross-plane segmentation.
In addition, the paper presents a data-partitioning-free learning approach to model more potential data distribution drifts, which is a strength of the paper. The paper addresses a question in medical domain, i.e., when one only has annotation on one plane, how to generalize the annotation to other planes, which is laudable goal.

**Strengths:**

This appears to be a novel contribution that "Unlike any previous prostate segmentation method or domain generalization approach, ADDG is based on the shape reconstruction method for 3D objects from single-planar images, so that it can estimate the 3D shapes of the prostate from single-planar MRI, enabling it to segment the prostate on unseen plane MR images better." such that the proposed work can estimate the boundary of prostate on the two unseen planes.

Other strengths include best performance in comparisons, and ablation study showing the effect of each component of the proposed method.

Another strength is the use of data-partitioning-free learning approach to model more potential data distribution drifts. This is a strength because it addresses an often-seen problem in medical imaging domain, i.e., data are often heterogeneous, especially if the data are from multiple sources.

**Limitations:**

Why do the authors say "multi-planar MRI uses almost the same scanning parameters but scans different internal structures, as shown in Figure 1 (f-h)", my understanding is that multi-planar MRI scans from different directions, but not different internal structures? Maybe the authors mean to say "different presentations of the internal structures" or something like that?

It is hard to understand that "The atypical domain differences between multi-planar MRI almost exclude features like image contrast", why is that? Are the authors implying that same organ may have different image contrast on different views of a multi-planar MRI?

"The combinatorial nature of polygons makes it possible to take derivatives in the space of possible meshes for any given surface. As a result, mesh processing and optimization techniques have difficulty utilizing the modular gradient descent component of modern optimization frameworks." Why is that? If one can take derivatives, then why is it difficult to use the modular gradient descent component? Authors need to elaborate on this.

The overall writing can be improved. Punctuation and appropriate use of blank space between, for example, Bongratz[5], should be taken care of.

Grammar can be improved, for example, "Due to the anisotropy of 3D MRI is reflected in intra-slice with high resolution and inter-slice with low resolution, we introduce a 2D boundary detection as an auxiliary task by designing an intra-plane" should be "Due to the fact that ..."
" as shown in 3"?

The paper seems to be prepared in a hurry as there are some errors in writing. For example,
"As shown in Figure 5 the subdivided mesh by Loop subdivision appears smoother compared to the subdivided mesh by midpoint", do you mean Figure 4? Otherwise, Figure 4 was never mentioned in the main text, and Figure 5 is MRI pictures, not meshes.
In equation (2) and (3) and the sentences around the two equations, what is the difference between p_k and p^k? Are they the same?
How are the \lambda's set in Equation (9)?

It is unclear how groundtruth was obtained in the experiments.

Why, in Table 1, does new method's Dice result, though the highest in mean value, have the highest variation among all the methods?
The section of data-partitioning-free meta learning is not well presented, a clearer presentation is recommended.

Regarding "We also crop the gray values to the 1st and 99th percentiles, then normalize them to [0,1] using Z-score normalization." Is this the 1st and 99th percentiles calculated separately on each plane or over all three planes? Or, are the percentiles calculated slice by slice?
Paper title in OpenReview is different from that in PDF.

**Suitability:**

2

---

### Official Review · Reviewer_3uwa · 2024-05-23

**Rating:** 5
**Confidence:** 4

**Summary:**

This paper introduces an interesting domain generalization setting, e.g, using axial planar scanned MRI for training, and then transferring to coronal and sagittal scanned MRI for testing, with data collected from different centers. The authors collected PROMISE12, NCI-ISBI13, I2CVB and Decathlon for training (axial plane), and tested the proposed ADDG on the PROSTATExSeg-HiRes dataset (coronal and sagittal plane). Although it appears novel, the method is overly complex.

**Strengths:**

1. A novel domain generalization setting.
2. Comprehensive experiments across existing prostate segmentation datasets.
3. Results indicate the effectiveness of ADDG under such a realistic scenario.

**Limitations:**

1. Considering the data size and common practice, a more meaning and challenge setting could be using coronal and sagittal plane scanned data for training, with generalization on axial planar scanned data.
2. Regarding to the complexity of model design, releasing the source code would contribute to the community.

**Suitability:**

2

---

### Official Review · Reviewer_YdtW · 2024-05-24

**Rating:** 4
**Confidence:** 2

**Summary:**

This paper aims to address the domain generalization problem in cross-plane MRI segmentation tasks. This problem arises due to the low inter-slice resolution of single-plane MRI, and unlike previous issues related to MRI data heterogeneity, the authors define it as a problem of discontinuous segmentation results. To tackle this issue, a 3D object surface reconstruction method based on single-plane images is introduced to better estimate the 3D organ shape. Additionally, a multi-path 2.5D segmentation branch is designed to handle data with varying degrees of anisotropy. Furthermore, an intra-plane prostate boundary segmentation branch is devised to leverage high-resolution 2D image slices from anisotropic MRI for learning 2D edge information of target organs. To enable the model to learn more domain-invariant features, an adaptive meta-learning strategy is proposed, making the data partitioning process of domain generalization learnable. The method achieves significant performance improvements in experiments, providing new insights and approaches for research and applications in related fields.

**Strengths:**

1.	This paper demonstrates strong innovation by identifying a previously unexplored domain generalization problem specific to prostate three-dimensional modeling, particularly in the context of cross-plane MRI segmentation.
2.	The authors introduce a novel meta-learning strategy that incorporates adaptive data partitioning for model training. This strategy allows for the full exploitation of the heterogeneity of multi-domain data and enables more comprehensive learning of domain-invariant features compared to traditional meta-learning methods.
3.	The methodology section of this paper is well-articulated, providing a clear and logically structured explanation of the proposed methods.

**Limitations:**

1.	Due to the involvement of initial data processing steps and the combination of multiple network structures, a thorough analysis of the algorithm's complexity is required. This may demand additional computational resources and time for both training and inference.
2.	For the boundary noise handling mentioned in the Intra-Plane Prostate Boundary Segmentation, the method utilizes first derivative operators to extract low-level boundary information. However, this may lead to sensitivity to noise, particularly given the various types of noise present in MRI images. Therefore, additional processing steps may be required to mitigate the impact of noise on boundary detection.
3.	In Data-Partitioning-Free Meta-Learning, the algorithm requires an initial model for training, which could potentially impact the performance of the final model. The paper opts for a structure similar to the 'Multi-Path 2.5D Segmentation Network', and it may be beneficial to provide some context here. If the initial model selection is inappropriate, it could affect the algorithm's convergence and generalization capabilities.
4.	When describing the experimental results, providing more detailed explanations would be beneficial. For instance, in the ablation experiments, it would be helpful to briefly explain why each component impacts the model performance.

**Suitability:**

3

---

### Official Review · Reviewer_5cZ4 · 2024-05-24

**Rating:** 4
**Confidence:** 2

**Summary:**

In this article, the author proposes an adaptive domain generalization framework for prostate cross-plane MRI segmentation. By introducing three-dimensional shape constraints of segmentation targets, the impact of information loss caused by the spacing between slices is significantly reduced. The adaptive data segmentation strategy also better elucidates the feature differences between different data planes.

**Strengths:**

The article proposes its own innovations to address the existing shortcomings in segmentation and demonstrates good performance on the corresponding data. The workload of the article is substantial, with a detailed description of the principles and structure. The graphics and logic of the article are good, making it a qualified level of work.

**Limitations:**

1.In section 3.2.1, the article describes the content of the Multi Path 2.5D Segmentation Network, which is used to process data with different degrees of anisotropy. However, the process of solving anisotropic data in this section is relatively vague and lacks detailed introduction.
2.In section 3.2.3, the author used a 3D triangular mesh deformation branch to produce the shape of target organization precision. What is the special feature of the emphasized triangular structure here?
3.In the description of Mesh Loss, the author's loss function contains many individual terms, all of which are assigned hyperparameters. Therefore, the relationship between them and the calculation of these hyperparameters can be described in more detail.
4.The author mentioned in the description of data preparation that first crop or fill to a size of 176x176x64, and then adjust it to 128x128x128. This processing method does not seem to be a simple cropping method, and adjusting the data in this way will affect some image information (especially for the information observed by doctors)
5.The description of comparative experiments and ablation experiments in the article is too limited. Given that many principles and methods have already been introduced, it is necessary to provide a detailed and accurate description of the experimental results to prove that the article's methods do have good effects

**Suitability:**

2

---

### Meta-Review · Area_Chair_RuVo · 2024-07-09

**Recommendation:** Accept (Poster)
**Confidence:** 4

**Metareview:**

The paper presents innovative contributions in addressing domain generalization challenges in prostate MRI segmentation, focusing on cross-plane variability and data heterogeneity. Reviewers generally acknowledge the paper's novelty. In the final revision, the authors need to provide more detailed insights into experimental results and analyses, such as computational complexity, and noise sensitivity.